# Anomaly Awareness

Charanjit K. Khosa[1] and Veronica Sanz[1,2]

[1]*Department of Physics and Astronomy, University of Sussex, Brighton BN1 9QH, UK*
[2]*Instituto de Física Corpuscular (IFIC), Universidad de Valencia-CSIC, Spain*

We present a new Machine Learning algorithm called Anomaly Awareness. By making our algorithm aware of the presence of a range of different anomalies, we improve its capability to detect anomalous events, even those it had not been exposed to. As an example of use, we apply this method to searches for new phenomena in the Large Hadron Collider. In particular, we analyze events with boosted jets where new physics could be hiding.

## I. INTRODUCTION

Algorithms that detect anomalies have to learn normal behaviour to be able to identify anomalous behaviour. Sometimes we do know what types of anomalies we need to search for, and then use supervised Machine Learning (ML) methods to find them. As anomalies are, by definition, rarer than normal events, these supervised techniques need to be adapted to unbalanced datasets and be made robust against fluctuations in the dominant *normal* or *in-distribution* dataset.

Oftentimes we do not know the whole set of possible anomalies we could encounter in data, or we cannot obtain a dataset with enough examples of anomalies. Supervised methods may perform well with known anomalies, but when applied to new ones they would typically not identify them. To design procedures to detect unknown anomalies, we then resort to unsupervised learning, trying to identify anomalies in a dataset as a function of e.g. of some form of *distance* within the dataset. This procedure is quite heuristic, often starting with a visualization of the data and some form of dimensional reduction, followed by some intuitive understanding of the problem. This is a hit-and-miss method, and in general the unsupervised strategies are substantially less powerful than a possible supervised method of detecting anomalies, see Ref. [1] for a review on novelty detection.

Here we present a different strategy, somewhat midway between supervised and unsupervised. We use the framework of a classification task (supervised learning) on a dataset with normal events, to introduce a concept of *awareness* of possible anomalies. We then use the output of the classification task to define a region where anomalies would concentrate.

We will show that this algorithm, after made aware of enough variety of anomalies, becomes effective at identifying more generic anomalies, even those the algorithm has not been previously made aware of. In other words, this anomaly awareness procedure becomes robust, i.e. more independent of the origin of the anomaly. In this sense our Anomaly Awareness algorithm is a hybrid method of learning, neither fully supervised nor unsupervised.

Note that there is a good body of literature in Computer Science proposing Deep Learning semi-supervised methods to detect anomalies in images, including medical applications. They are mostly based on AutoEncoders and GANs, see e.g. Refs. [2–6]. In this context, Anomaly Awareness is a new type of semi-supervised procedure, applicable to input images but also to other types of information.

We will exemplify the use of this method in a non-trivial task in Particle Physics. In the context of the Large Hadron Collider (LHC) searches for new phenomena, we show how Anomaly Awareness can help making these searches more robust, less dependent on the specific scenario one has in mind. This *model-independence* of LHC searches is particularly important now that the traditional ways of thinking in Particle Physics are challenged by the absence of expected discoveries at the LHC.

Note that a handful of recent studies [7–23] proposing ML algorithms to perform model independent searches [1] are showing impressive reach for the considered toy examples.

This paper is organised as follows. In Sec. II we describe the general algorithm of Anomaly Awareness as well as the Convolutional Neural Network (CNN) architecture we have employed for the subsequent analysis. We show how we use it in Particle Physics in Sec. III and then conclude in Sec. IV.

## II. ALGORITHM DESCRIPTION

We now explain in detail the Anomaly Awareness algorithm, represented in the Algorithm Description 1. The starting point of the algorithm is a classification task which in its simplest form is a binary classification task.

In this initial run (*prior run*), the algorithm learns to classify only *normal* classes, and is not yet aware of the presence of anomalies. The end result of this run would be a trained algorithm with some choice of optimal hyper-parameters, which will be used to initialize the next run, the *anomaly awareness run.*

In the second run the algorithm will now see some anomalies. The new loss function contains the same term as in the prior run, e.g. cross-entropy for a binary classification task, but has a new term (Anomaly Awareness) which distributes the anomalous samples uniformly across the classes, e.g. assigning 50% probability of belonging to each class in a binary task, or $1/n$ probability in a $n$-class baseline classification task. In case of binary classification, anomaly loss term ($l_2$)

[1] Other than the model independent searches (in a unsupervised fashion) recent studies also proposed the use of ML methods for specific tasks to extract maximum information from Particle Physics experiments data, see e.g. [24]).

has a following form for each data point

$$l_2 = -\left(\frac{p_1 + p_2}{2} - \log(e^{p_1} + e^{p_2})\right) \qquad (1)$$

where $p_1$ and $p_2$ are the predicted probabilities by the (normal) binary classifier[2].

The Anomaly Awareness term is modulated by a parameter $\lambda_{AA}$, which sets the relative importance of anomalous examples respect to the normal samples in the loss function.

So far this algorithm is similar to Outlier Exposure [25], but in our case the AA term will contain an array of different anomalies which, as we will show later, is crucial to allow the algorithm to detect unknown anomalies. Another component of Anomaly Awareness, not present in Outlier Exposure, is the use of the classifier output $p$ to obtain an optimal window $[p_{An}^{min},\ p_{An}^{max}]$ to detect anomalies over a large background of normal events.

**Algorithm 1** Anomaly Awareness (AA). Important parameters are $\lambda_{AA}$, $p_{An}^{min}$, $p_{An}^{max}$.

**Prior Run**

Initialize test:train splitting of *Normal* ($N$) dataset
Initialize hyper parameters
Initialize Model (CNN architecture)
**for** Training over the epochs **do**
  Cross entropy loss
  Update model parameters.
**end for**
Get accuracy for $D_{test}$ and $D_{train}$
This run sets the hyper-parameters for the AA run

**Anomaly Detection Run**

Load the *Anomaly* ($An$) dataset
Initialize amount of data w.r.t. the *Normal* dataset
Initialize $\lambda_{AA}$
**for** Training over the epochs **do**
  $l_1$ = Cross entropy loss (*Normal* dataset)
  $l_2$ = Cross entropy loss (*Anomaly* dataset with Uniform Distribution)
  Loss $= l_1 + \lambda_{AA} l_2$
  Update model parameters.
**end for**
Get softmax probabilities for all the datasets,
$p_i,\ i = N,\ An$
Select datapoints in a range $[p_{An}^{min},\ p_{An}^{max}]$,
range optimized to select anomaly over normal events

In the application to LHC searches of Sec. III the input to this analysis will be in the form of 2D images of jet spatial structure, hence the neural network architecture is made of a few convolutional layers and a final classification layer, see Fig. 1 for the specific choice we made.

## III. A NON-TRIVIAL EXAMPLE OF ANOMALY AWARENESS: BOOSTED HADRONIC PHENOMENA

[2] Note that we use softmax probability for the predicted label in cross-entropy and write here the expanded expression

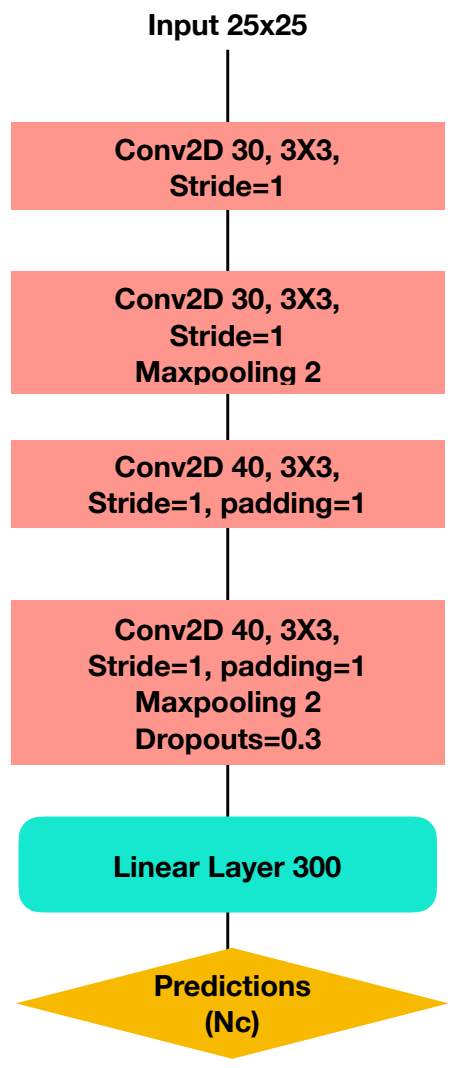

FIG. 1. CNN Architecture used in this study. Input images are passed through a set of convolutional layers and end in a linear layer providing predictions for the classification problem.

We now describe how this algorithm could be used in the area of Particle Physics, in particular in searches for new phenomena at the LHC. This description and the results we provide exemplify a *use case* in our field domain, but are not to be taken as the optimal procedure to follow in a realistic analysis. The LHC's environment is very complex, and modelling the behaviour of collisions requires a sophisticated machinery which we are just approximating here.

Let us first motivate the problem we want to tackle. The aim of Particle Physics is to understand the Laws of Nature at the most fundamental level. These laws do take very simple forms when described in terms of the right mathematical objects, but in terms of empirical probes they adopt tremendously complicated manifestations.

A perfect example of this complexity is the LHC, one of the best probes of Nature we currently have, where massive amounts of data are collected and analysed to test the so-called Standard Model (SM) of Particle Physics.

At the LHC, collision data is transformed into many measurements of SM observables, providing precise tests (per-cent and even per-mille precision) of the validity of the SM as a paradigm to explain Nature.

So far the SM has passed all these tests with flying colours. Yet we know the SM paradigm, albeit very successful, is not complete. The SM does not explain the Universe as we see it, where 95% is made of dark stuff (Dark Energy and Dark Matter) the SM does not account for, and for the rest 5% we do not understand how antimatter got out of the way, or how neutrinos got massive.

Thus to answer the question *'How does the Universe work?'* many experiments are looking for ways to find new phenomena, beyond the SM framework. At the LHC these searches take many forms, and in this example we are going to focus on certain types events where high energy jets are produced and new phenomena (beyond SM) could be found.

The SM interactions do produce these jets, for example in the form of quarks and gluons which then hadronise in the detector. We will denote these *normal* events as two classes: *Top* and *QCD*, and later add a third SM class, *W-jet*.

Searches are focused on finding some anomaly in the behaviour of these jets which would indicate the presence of a new set of laws at play. We simulate anomalies produced by new particles, which we denote as *resonances* leading to jets with 2-, 3- or 4-prongs ($R_{2,3,4}$), or new effective interactions which we denote as *EFT* [3].

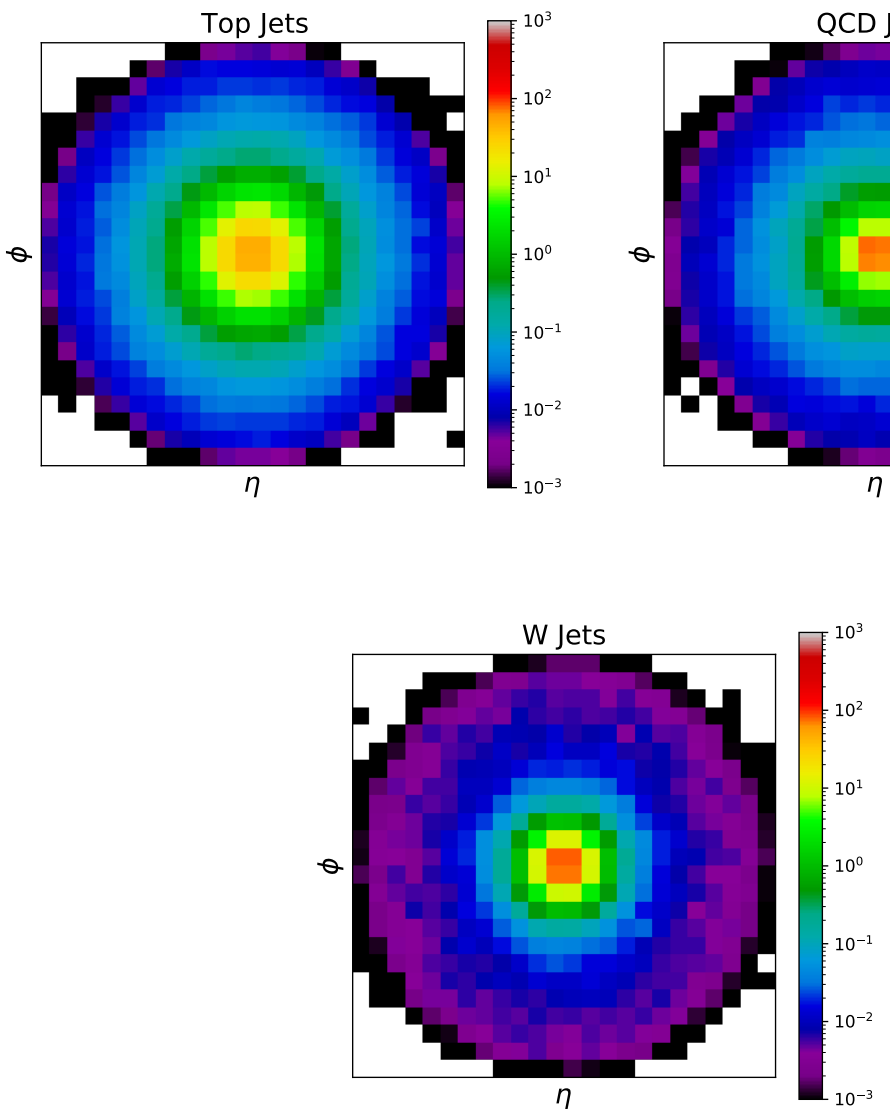

FIG. 2. Average jet images for SM processes. From top to bottom and left to right: Top, QCD and W-jet.

### A. The input information

To study anomalies in these events we will represent them as follows: all the information on directionality, timing and energy deposition of the event is reduced to gathering the largest amount of energy collected around a cluster (*leading fat jet*) in the hadronic calorimeter of the detector. We then represent the angular distribution ($\eta$, $\phi$) of the energy depositions inside the fat jet with a color coding that encodes relative amounts of energy (in GeV). The typical distributions for these events are shown in Figures 2 and 3, where we see the differences among the sources of SM fat jets and possible new phenomena. These images are an average of all the events we have simulated ($\sim$ 50K events per scenario) and one should note there is substantial variability among events from the same source [4].

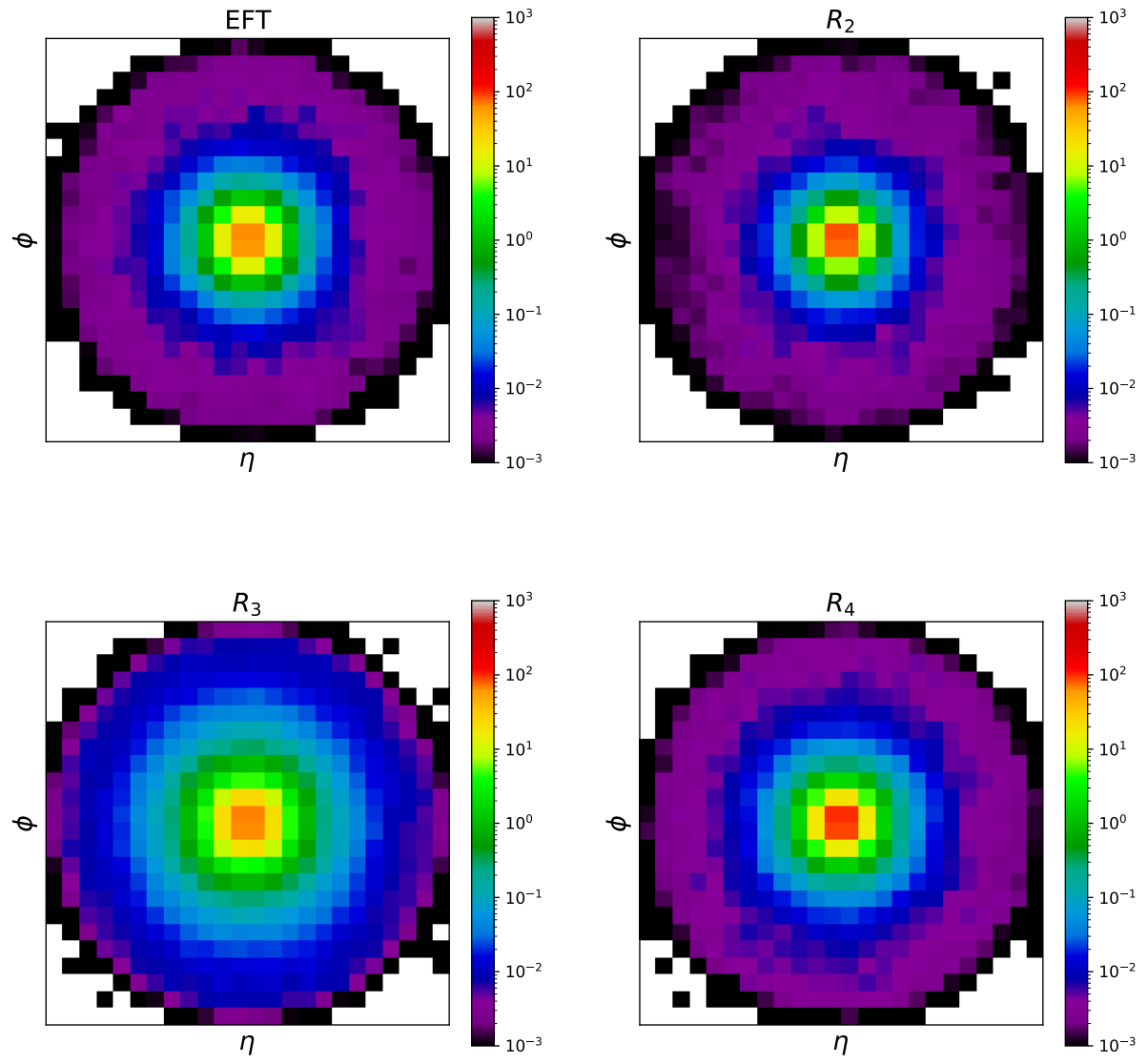

FIG. 3. Average jet images for New Physics processes. From top to bottom and left to right: EFT, resonance into boosted $ZZ$, resonance into boosted tops, and resonance into boosted pairs of Higgs bosons.

### B. The prior run

Using these images as input, we run an initial classification task of two classes in the *normal* distribution (Top *vs* QCD) using the convolutional architecture in Figure 1 with batch size 100, 100 epochs and ReLU as activation function in all the layers [5].

Once trained over examples of Top and QCD events, the algorithm would give a prediction event-by-event of the probability of belonging to the class Top or class QCD. If we output the predictions for true Top and QCD events , e.g. in terms of Top probability, a good algorithm should distribute Top events near 1 and QCD events near 0, leading to two sharp peaks of the probability distribution function (PDF).

But what about any other types of events? We can also run the algorithm over anomalous examples and see where these are distributed in the Top probability axis. This is shown in Figure 4, where observe that other scenarios would typically be misclassified as Top

---

[3] EFT interactions correspond to Higgs production in association with a Z-boson as described in Ref. [26] and switching on a coefficient $c_{HW}$ as defined in Ref. [27] within the limits obtained in Ref. [28]. The $R_{2,3,4}$ examples were generated with a RS model decaying into boosted $ZZ$, boosted $t\bar{t}$ and $hh$ with $h \to 4$ jet, respectively.

[4] These images have been produced by running Monte Carlo simulations of 13 TeV LHC collisions at parton-level with aMC@NLO [29, 30] and then showering and hadronizing with Pythia [31, 32]. We then used Pythia 8 SlowJet program for clustering. The main cuts applied to the events were finding a leading anti-$k_T$ jet of $R = 1$, $p_T > 750$ GeV and $m_J \in$ [50,300] GeV.

[5] Note that CNNs have been used for the Top *vs* QCD jet classification problem in Refs. [33, 34]. Also note that some works are using ML techniques to improve on the task of top tagging, e.g. Refs. [35, 36], which could be incorporated to the initial AA run.

or QCD. In other words, the classification task specializes on Top and QCD events characteristics, and any new type of scenario, which could exhibit other event characteristics, is mostly placed into one of the two classes. These *anomalies* are mis-identified as *normal* classes, QCD or Top.

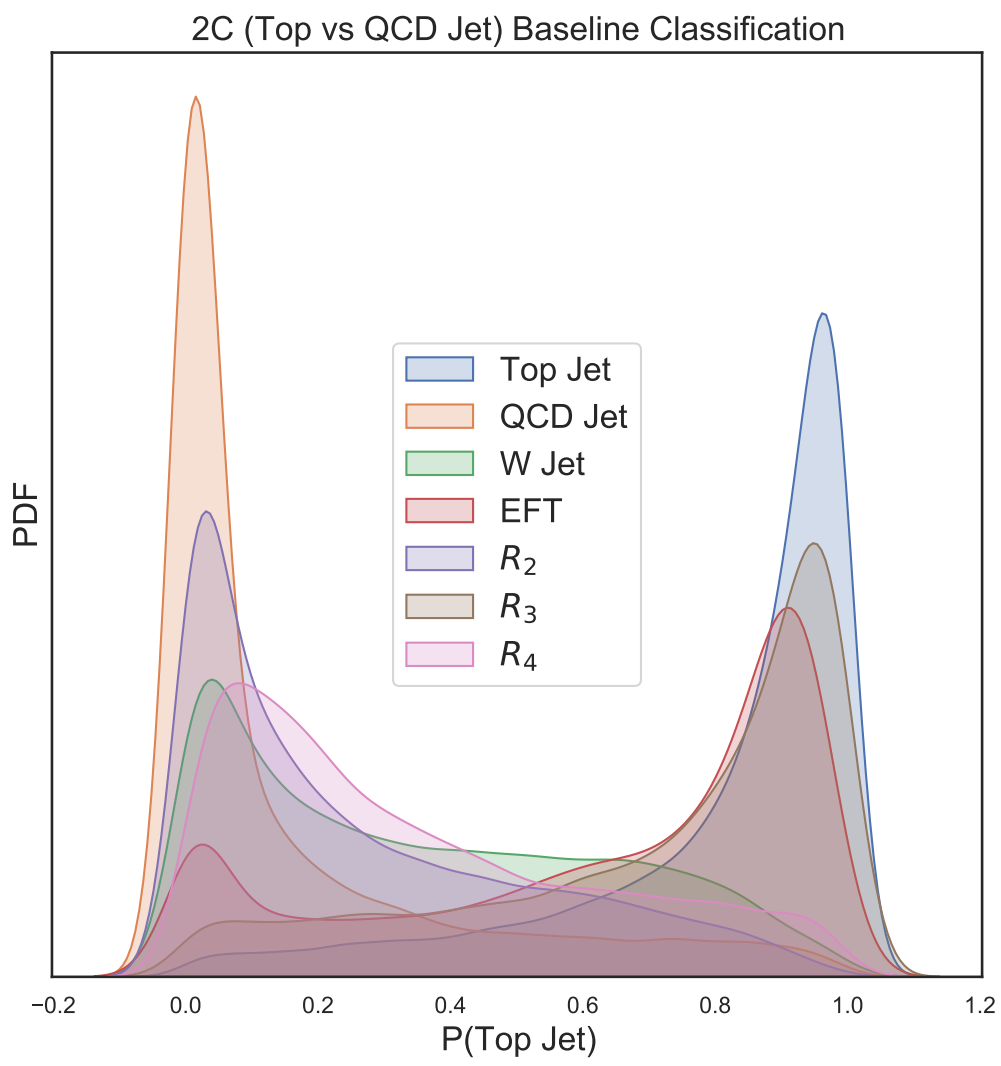

FIG. 4. Output of the binary classifier (Top *vs* QCD) on events from different sources (Top, QCD, W-jet, EFT and Resonances).

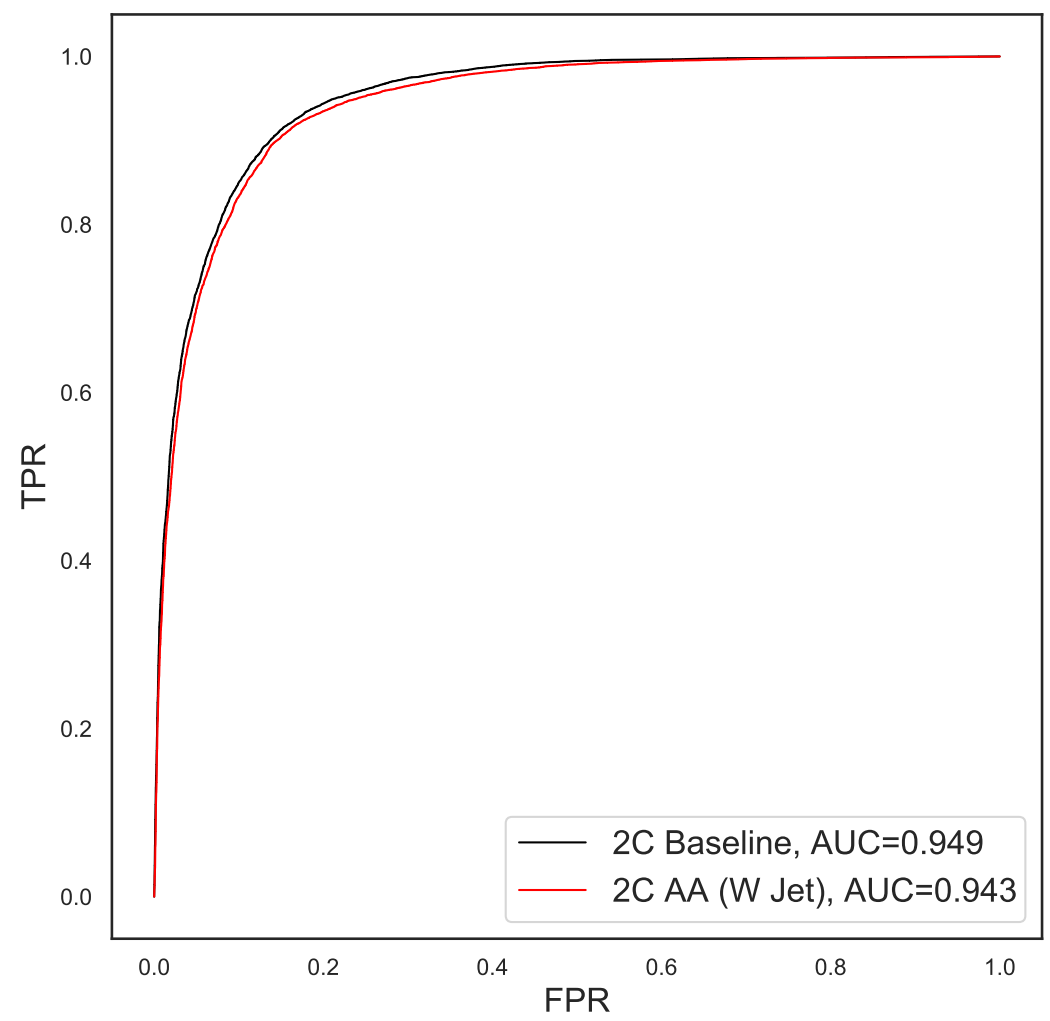

FIG. 5. ROC classification curve for the prior run, and for a run including AA of W-jet. Similar curves are obtained when adding more AA types.

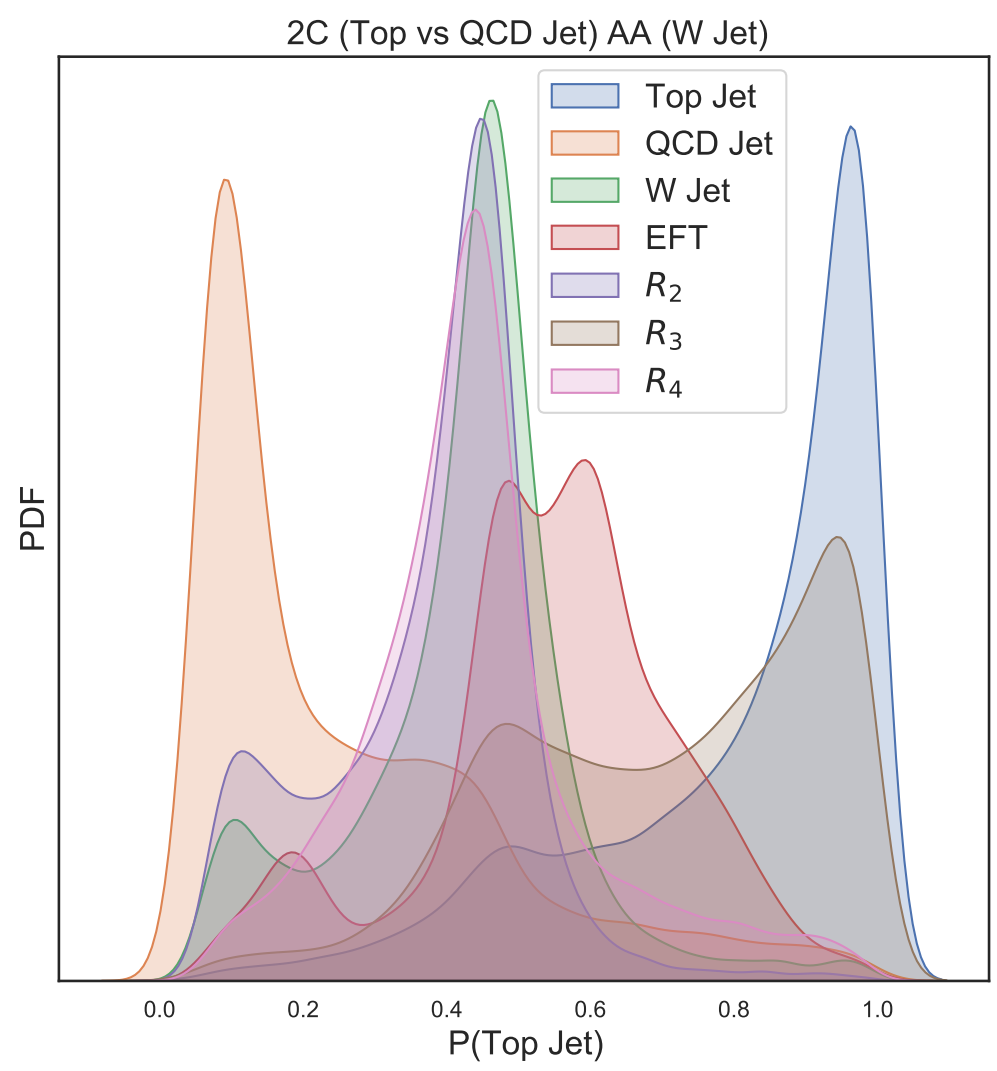

FIG. 6. Output of the Anomaly Awareness binary classifier (Top *vs* QCD) on events from different sources (Top, QCD, W-jet, EFT and Resonances). An Anomaly Awareness term has been included with only W-jets.

### C. Adding Anomaly Awareness

Now let us introduce Anomaly Awareness and, for the moment, just introduce awareness to a single new type of events, W-jets. In terms of the classification results, the effect of adding an AA term is not substantial, see Fig. 5 where we see how the ROC curves with and without AA terms are essentially identical. We checked that this result does not change when adding more AA terms.

However the effect on the anomalies, all of them, is substantial. As the algorithm becomes aware of possible anomalies, even when exposed to only one type, it does also become better at separating QCD/Top from other types. This is shown in Fig. 6, where now the probability distribution of anomalies gathers towards the centre of the distribution, i.e. they are classified neither as Top nor as QCD.

As one sees in Fig. 6, events with P(Top) close to 1 are mostly coming from a Top distribution and those whose P(Top) is close to 0 are mostly from the QCD events. One could think on using this behaviour to assign an *anomaly* character to events which would be well separated from the Top and QCD, e.g. whose P(Top) would lay near 0.5. But as we will see later in Sec. III E, this definition would be too naïve for Particle Physics purposes. Indeed, in reality in a sample of LHC events there would be a variable number of QCD and Top events depending where on the ROC curve we are setting our analysis. Moving towards the right on Figure 6 corresponds to different choices of working points in the acceptance of Top and QCD events, an *efficiency* to collect or reject these events. Looking at a rough window around P(Top)$\simeq$ 0.5 does not take into account the overall amount of QCD and Top events remaining after setting the threshold. We will discuss this issue in the last part of this Section.

As we introduce awareness to more types of anomalies, this behaviour continues to hold and improves up to a point. This can be seen in the Figures in 7. In the top panel we observe the effect of adding an additional example in the awareness term, adding to W-jet additional awareness of $R_4$, a resonance leading to a high energy jet with 4-prongs. The improvement from Figure 6 is clear, signalling that the awareness procedure

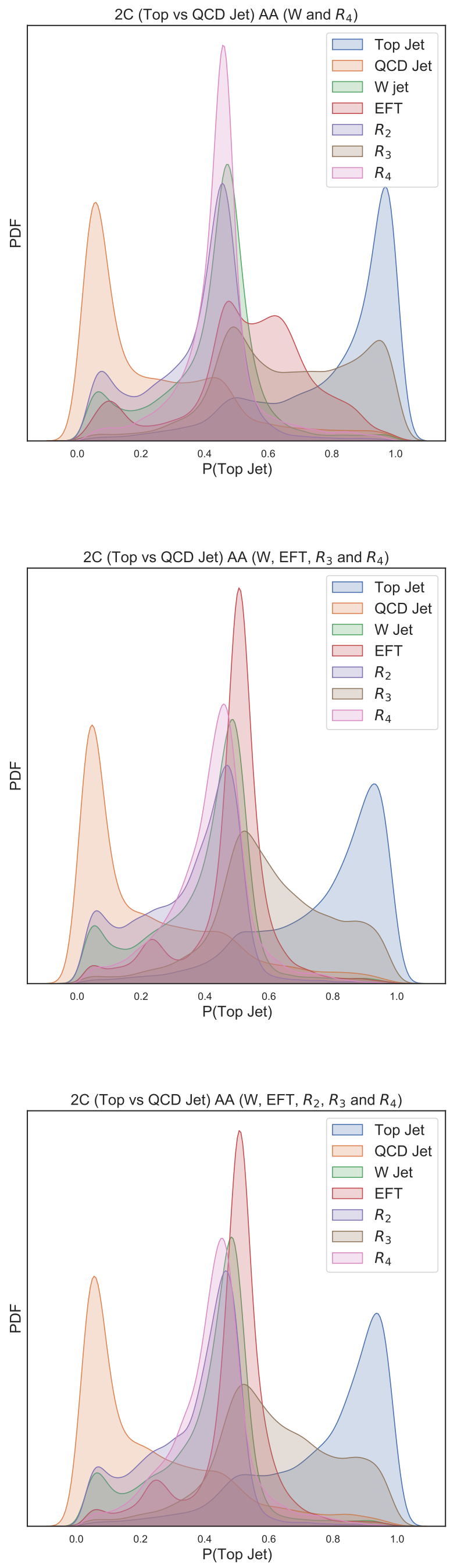

FIG. 7. Output of the binary classification (Top *vs* QCD) on events from different sources (Top, QCD, W-jet, EFT and Resonances). Different Anomaly Awareness terms have been included: W-jet and $R_4$, a resonance leading to a high energy jet with 4 prongs (*top figure*), plus EFT and $R_3$, a resonance leading to a jet with 3 prongs (*bottom-left figure*) and finally adding to the former $R_2$, a resonance leading to a jet with 2 prongs (*bottom-right figure*).

improves with more variety of examples. We checked that the improvement is roughly independent of the choice of examples of anomalies, which indicates the procedure is robust.

Nevertheless, this improvement does not imply that awareness can be arbitrarily enhanced by just adding more examples. Indeed, we find that the power of the procedure *saturates*. This can be seen in the bottom panels of Fig. 7, where going from awareness of four different anomalies to extending to five does not change the overall picture.

This saturation is to be expected: the amount of information in the images we created is limited intrinsically and by design, as we are selecting just the leading jet in the event and plotting only angular distributions of energy depositions. Some additional information could be added to the analysis, as even in that leading jet one could add more information, like probability of the presence of a b-jet. And beyond the leading jet, important correlations with the other parts of the event could be added in this analysis. Hence we would expect a more detailed analysis to lead to better performance, although this is not the main focus of this work, which is presenting the idea of Anomaly Awareness and how it would qualitatively work in an LHC set-up.

Let us finish discussing the effect of the modulation term $\lambda_{AA}$. This term sets the relative importance of normal examples shown to the algorithm, in the cross-entropy function, versus the number of anomalous examples subject to a uniform distribution. We can think on two limiting cases. On one hand, a very small value of $\lambda_{AA}$ would lead to the same result as the prior run, and would not bring the anomalies to the centre of the classification output. On the other hand, a large value of this parameter would degrade the prior classification task, broadening the PDFs for Top and QCD, the backgrounds we are fighting against. Somewhere in between, with a moderate amount of awareness, the optimal performance lies. In this work we have used a near-optimal value ($\lambda_{AA} = 0.5$). In the Appendix we discuss the comparison with other choices of $\lambda_{AA}$.

### D. Generalization to more than two categories

So far we have shown results based on a binary classification problem (Top *vs* QCD), but Anomaly Awareness could be generalized to classification problems with more than two classes. The only difference in the algorithm 1 would be in the AA term, where the Uniform Distribution would be along all the classes. To illustrate this procedure, we repeat the analysis, now with three SM classes: Top, QCD and W-jet.

After training with a *normal* dataset with equal amounts of Top, QCD and W-jet, the algorithm can provide for each new event a probability of belonging to each class. In Figure 8 we represent the PDF of events within these three categories (P(Top Jet), P(QCD), P(W-jet)). True top events (in red) are mostly gathered around values of one for P(Top Jet) and zero for P(W-jet) and P(QCD). Similarly true W-jet events (green) gather around values of one for P(W-jet) and zero for the others. This plot is 2D, but if we had plotted P(QCD), we would observe a similar behaviour: most true QCD events would be correctly classified.

As in the two-class case discussed before, the prior classification algorithm, when faced by new types of events, would likely misclassify them as one of the known categories. For example, EFT anomalies would be mainly misclassified as W-jets. This is shown in

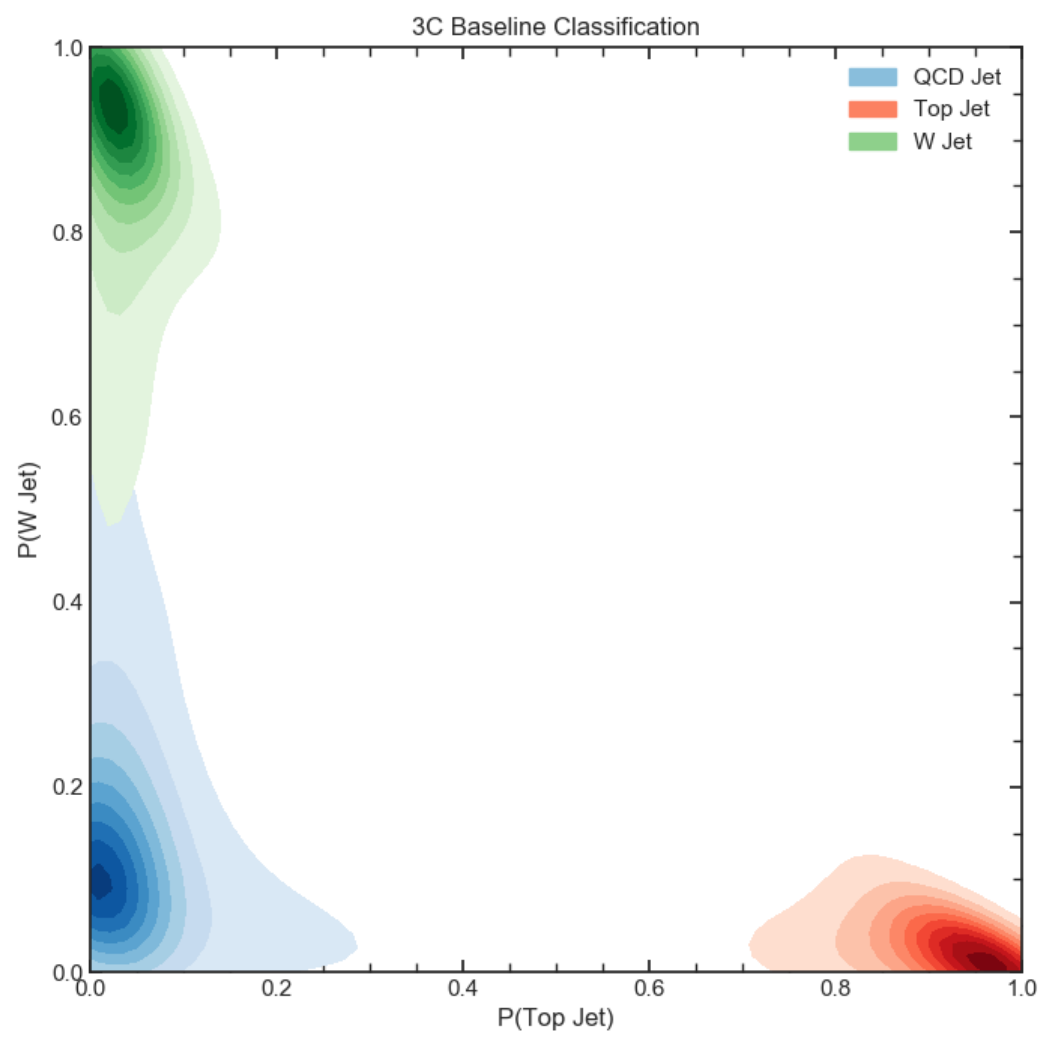

FIG. 8. Probability distributions for the *normal* classes in the three-class example: Top (red), QCD (blue) and W (green). The axes are the probability of an event to belong to class Top Jet (x-axis) or W Jet (y-axis).

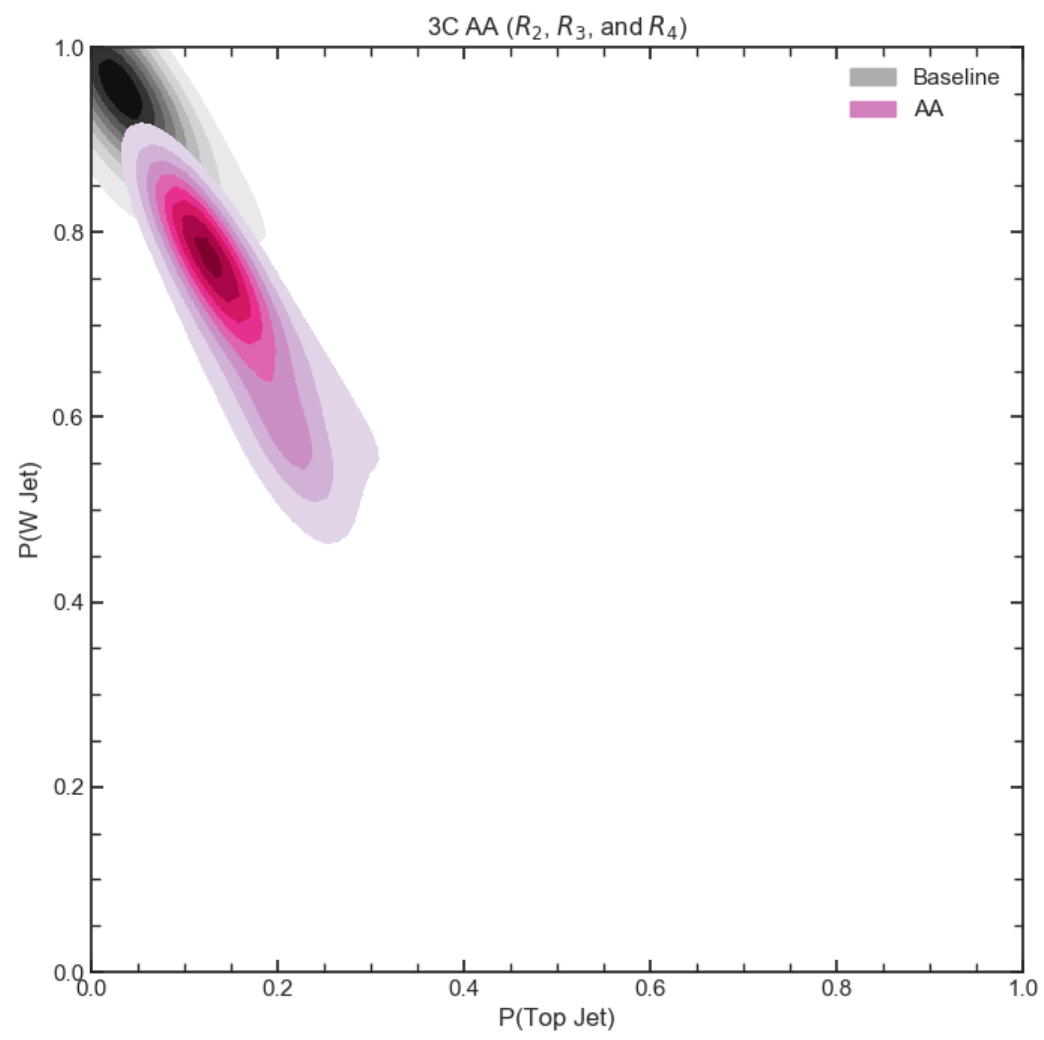

FIG. 9. Probability distribution of EFT events after the prior run (black), and the effect of Anomaly Awareness on the distribution of EFT events (pink), when the algorithm is made aware of all the anomaly classes except EFT. Axes are the same as in Fig. 8.

Fig. 9, where the black distribution represents the PDF of EFT events previous to introducing AA.

If we then run the model with Anomaly Awareness of all the anomalies discussed before (except EFT), the EFT events move towards the center of the PDF plane, represented by the pink blob in Fig. 9. In other words, despite not being aware of EFT-type anomalies, exposure to other anomalies does help separating EFT fat jets from SM sources. We checked that adding EFT to the AA term on top of the other cases do not change this picture qualitatively, again indicating a *saturation* of the amount of information in these events which seems to be already covered by the diversity of $R_2$, $R_3$ and $R_4$.

### E. Anomaly detection

So far we have discussed the effect of AA in the classification task. We noted that AA preserves the performance of the classification task respect to the prior run, which is to identify correctly *normal* classes. This can be seen from the comparison of ROC curves in Fig. 5, where the overall effect of adding AA terms is negligible. But the effect on the anomalous events is substantial, bringing the distribution of predictions for anomalous events farther from the region of the *normal* classes, which gather around 0 and 1, see Figs. 6 and 7.

Now we want to discuss how this separation could be used in practice in an LHC search for anomalies. Note, though, that the following quantitative discussion is intended for illustration and not to be taken as a full-blown analysis of anomalies in high-$p_T$ fat jets at the LHC. As mentioned before, the LHC environment is complex, and modelling the behaviour of hadronic final states requires a sophisticated machinery which we are just approximating with simple theoretical simulation tools. Moreover, we have only considered information on the leading jet, missing then important correlations with other hadronic activity or correlated channels.

With all these caveats in mind, we describe a procedure one could follow to use AA in order to increase detection of anomalies.

To claim an anomaly detection we need a statistical criteria to determine how many anomalous events ($N_{An}$) over SM events $N_{SM}$ are required. A typical criteria used in Particle Physics is:

$$\mathcal{S} = N_{An}/\sqrt{N_{SM}}, \tag{2}$$

where $\mathcal{S}$ indicates statistical significance and one can choose a value, $N_{An}/\sqrt{N_{SM}} = 5$, as a benchmark to claim the significance of the anomalies is above statistical fluctuations in the SM background with a 5-$\sigma$ confidence level. This criteria is commonly used in Particle Physics, see e.g. Ref. [37] for a typical use of this criteria to claim discovery or exclusion of new phenomena. Also it is worth noticing that that the criteria cited in Eq. 2 is an asymptotic limit of a more robust statistical treatment for a large number of events, see Ref. [38] for more details.

The number of events $N_{An,\,SM}$ depends on how often these types of events are produced in LHC collisions, i.e. the cross-sections $\sigma_{An,\,SM}$ [6]. It also depends on the thresholds we choose when applying the algorithm, i.e. how many of these events we reject and collect.

---

[6] A cross section is a probability of interaction per unit area and at LHC collisions the number of observed events is related to the cross section via the data amount collected, also know as luminosity $\mathcal{L}$.

In Figures 6 and 7, one could choose such criteria as a window in the output probability of the classifier

$$p \in [p^{min}, p^{max}] \tag{3}$$

and scan different windows to obtain the maximum efficiency to collect anomalies and reject SM events.

The effect of this scan is shown in Fig. 10, where we plot the following quantity

$$R = \frac{\epsilon_{An}}{\sqrt{\sigma_{QCD}\,\epsilon_{QCD} + \sigma_{t\bar{t}}\,\epsilon_{t\bar{t}}}}. \tag{4}$$

In this equation $\epsilon$ denotes the area of the PDF curve in Figures 6 and 7 on a given window.

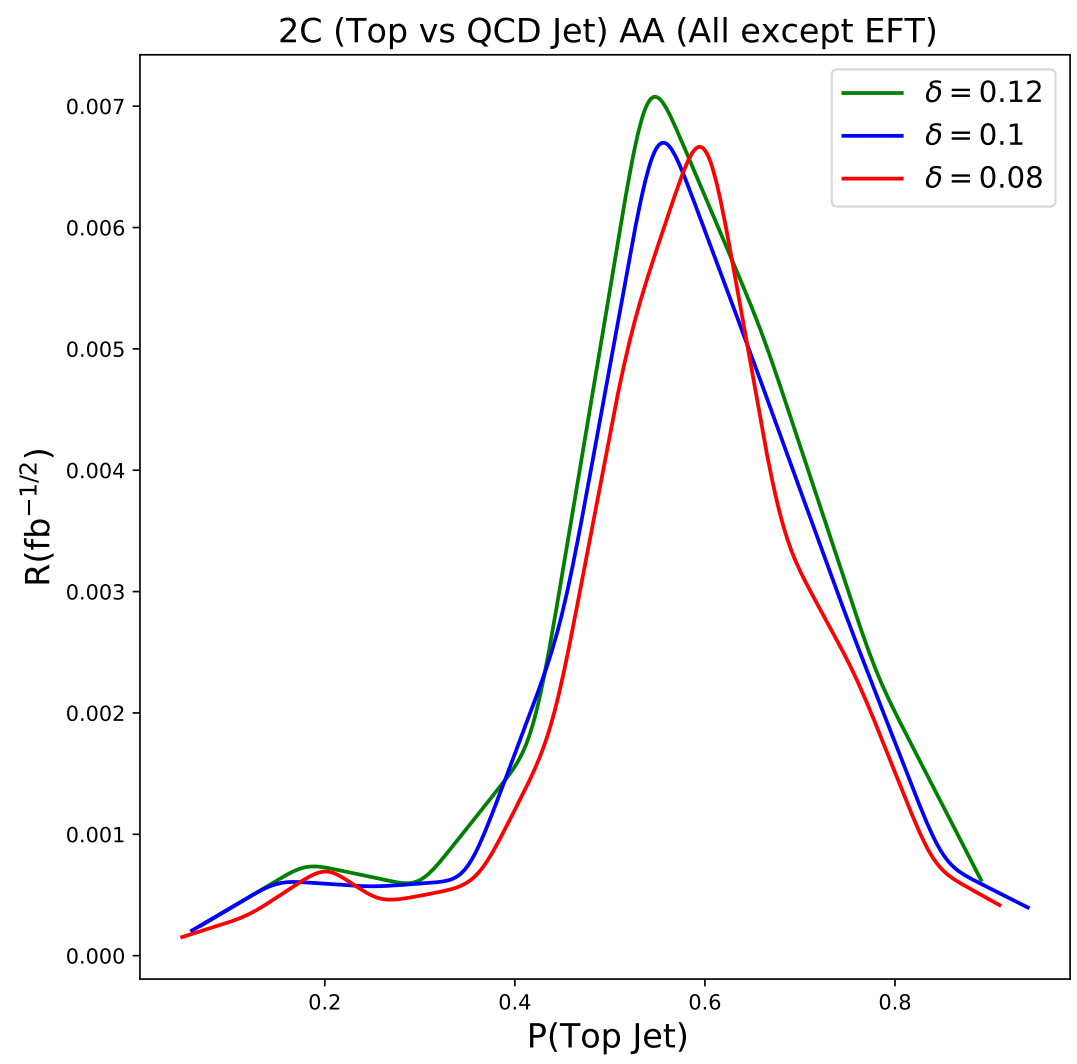

FIG. 10. Value of $R$ defined in Eq. 4 as a function of the EFT output classifier P(Top Jet) of an AA run with awareness of all the anomalies except EFT. The three curves correspond to window widths $\delta = 0.1$, 0.08 and 0.12.

Note how in $R$ the QCD and Top cross-sections are weighted in. Right after the high-$p_T$ selection cuts, the QCD total cross section is much larger than the Top. But one can use the output of the classifier to impose a threshold on P(Top Jet), and drastically reduce the amount of QCD events, closer to the amount of Top events.

In anomaly detection, the task of identifying anomalies means fighting against QCD and/or Top, depending on where in the output classifier region our window lies. Towards the left, P(Top Jet) $\ll 1$, QCD is the dominant contribution to the denominator in $R$, and at the other end, Top is dominant. Somewhere in between these two extremes we should find the best window for anomaly detection. In Fig. 10 we see exactly that behaviour. $R$ is very small on both ends of the plot, where the QCD and Top backgrounds are overwhelming. As we move our window $[p^{min}, p^{max}]$ towards the center, both QCD and Top drop. Therefore, at the maximum of $R$, $R_{max}$, the statistical criteria 2. The parameter $\delta$ in this plot corresponds to the width of the window, $\delta = p^{max} - p^{min}$ and one can see the value of $R_{max}$ does not depend strongly on the choice of $\delta$ as long as it is $\approx 0.1$.

After determining $R_{max}$, one can turn the criteria for discovery $N_{An}/\sqrt{N_{SM}} = 5$ into a minimum value of the anomaly cross section one would be able to detect. This value would depend on the amount of data collected at the LHC (i.e. luminosity, $\mathcal{L}$), hence on the time it runs. Indeed, note that

$$\frac{N_{An}}{\sqrt{N_{SM}}} = R\,\sigma_{An}\,\sqrt{\mathcal{L}} \tag{5}$$

hence

$$\sigma_{An}^{min} = \frac{5}{R_{max}\,\sqrt{\mathcal{L}}} \tag{6}$$

which is shown in Fig. 11 for the EFT case (where AA is to all anomalies but EFT). We repeated the same analysis for other anomalies, $R_2$, $R_3$ and $R_4$, with similar results, as expected from a procedure which aims to achieve robustness and model-independence.

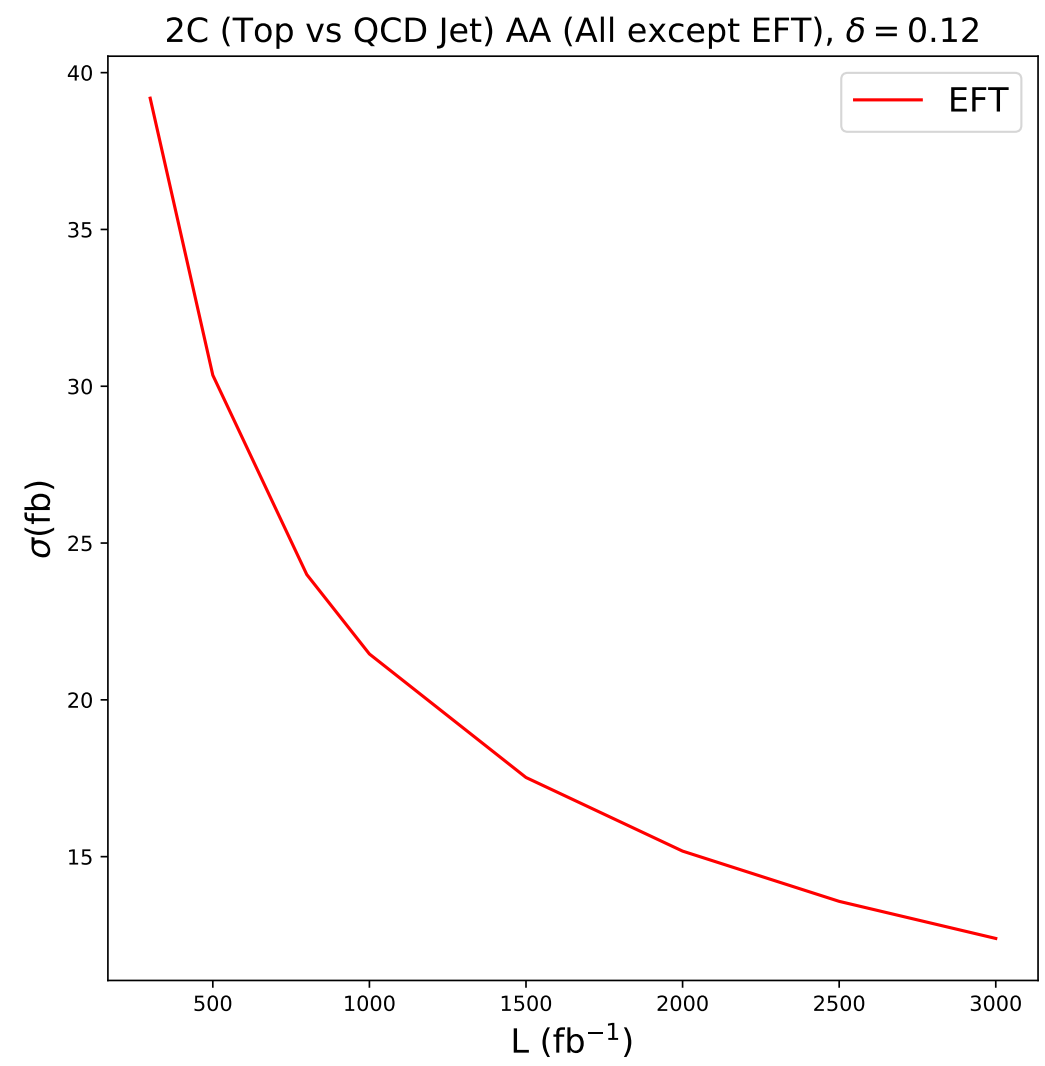

FIG. 11. Value of $\sigma_{An}^{min}$ (in fb) as a function of LHC luminosity (in fb$^{-1}$). The value of 3000 fb$^{-1}$ corresponds to the expected luminosity of the HL-LHC run.

As a reference, the QCD cross section after the selection cuts is of the order of $50\times10^3$ fb, and Fig. 11 shows that at High-Luminosity LHC ($\mathcal{L}$ =3000 fb$^{-1}$) we would be able to detect cross sections for anomalies of the order of 10 fb, a 1:5000 ratio of anomaly over in-distribution.

We do not want to finish this section without stressing once more that the results shown in Figs. 10 and 11 should be taken as a qualitative illustration on how to use AA for anomaly detection. A better simulation and analysis, including more information on the events and more types of anomalies, would likely lead to substantially better results than those shown here.

## IV. CONCLUSIONS

In this paper we have described a new method of anomaly detection, based on a classification task

within a multiclass in-distribution, and the effect of adding to the task some level of anomaly awareness. This is a semi-supervised method of anomaly detection, where the algorithm is exposed to a variety of anomalies to render it less dependent on specific anomalies.

As a use case for this method we have addressed a non-trivial task of anomaly detection in Particle Physics. Using information of LHC events we have studied how Anomaly Awareness can help to establish a more model-independent strategy to search for new phenomena at high energies. We observe that Anomaly Awareness does not substantially interfere with the underlying classification task, see Fig. 5, but brings the anomalies to a region where a separation is possible, see e.g. Figs. 4 and 6. We illustrated the use of this separation in Sec. III E.

We found that awareness of *any* anomaly helped on detecting others, and that adding more anomalies to the AA term did improve detection of new, unknown situations. We did notice, though, that this procedure levels off after awareness to a few examples, likely to indicate that the feature extraction ability of the algorithm has saturated.

Although we constructed jet images as input for the algorithm, Anomaly Awareness could be used with any type of input. For example, for the LHC application we could have used instead images of the leading and subleading jets simply pasted together, as proposed in Ref. [39], event information terms of a set of kinematic variables, mixed input, or even lower-level event information (closer to the raw output of the detector). To illustrate the use of AA in other situations, in the Appendices we show the application of this method to a non fully hadronic LHC final state and to a non-physics dataset.

Finally, our discussion on LHC anomaly detection should be understood as a *proof-of-concept* on the use of Anomaly Awareness, and not as a dedicated study. We nevertheless find promising results, despite using just a part of the information available in the LHC events. And although we showed results with the EFT as the unseen anomaly, we found similar results for the other anomaly examples. Compared with supervised ML methods for EFTs [26], we find that our estimative limit of the anomalous cross section, Fig. 11, is of the same order of magnitude and motivates a more systematic study.

## ACKNOWLEDGEMENTS

CKK is supported by the Newton Fellowship programme at the Royal Society (UK). VS acknowledges support from the UK Science and Technology Facilities Council (grant number ST/L000504/1), the GVA project PROMETEO/2019/083, as well as the national grant FPA2017-85985-P. The authors gratefully acknowledge the computer resources at Artemisa and the technical support provided by the Instituto de Fisica Corpuscular, IFIC (CSIC-UV). Artemisa is co-funded by the European Union through the 2014-2020 ERDF Operative Programme of Comunitat Valenciana, project IDIFEDER/2018/048.

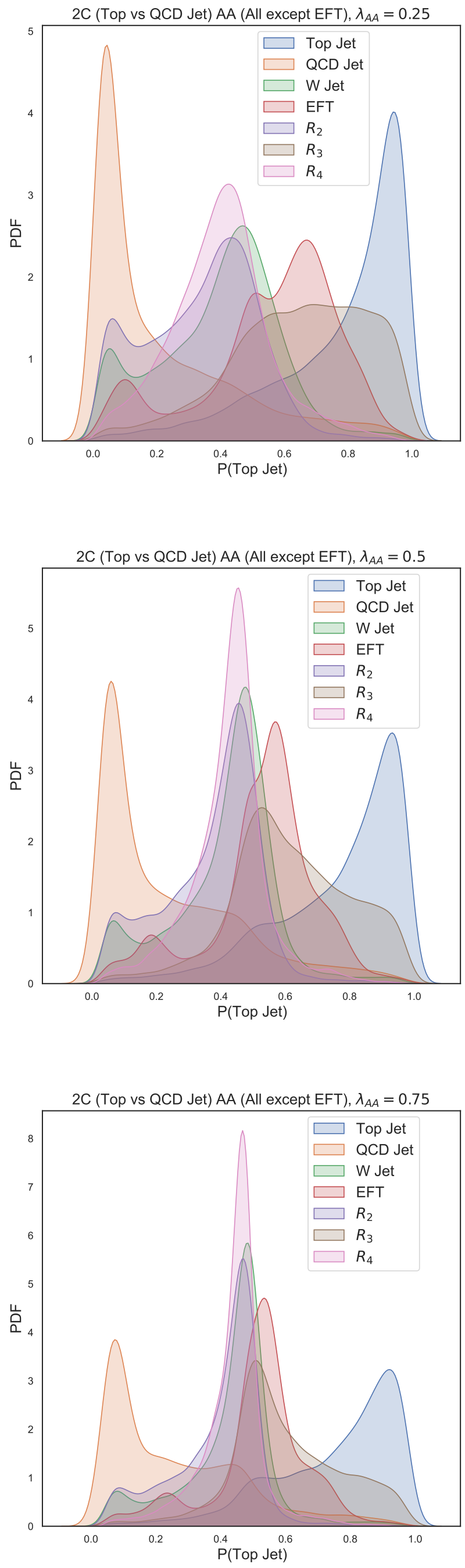

FIG. 12. Output of the binary classification (Top *vs* QCD) on events from different sources (Top, QCD, W-jet, EFT and Resonances) with AA (All except EFT) for different $\lambda_{AA}$ values.

## Appendix A: Choice of $\lambda_{AA}$

In this appendix we show the impact choice of the hyper-parameter $\lambda_{AA}$ made in the Anomaly Awareness run with the LHC fat jets example, although similar results are obtained for the other two examples shown in this paper. In Fig. 12 we see how, as explained in the main text, increasing the value of $\lambda_{AA}$ does produce a larger density of anomalies in the central region, but at the cost of degrading the baseline task. In the second part of the algorithm we choose a window in the central region where the amount of

normal versus anomaly is as low as possible, hence our choice of $\lambda_{AA}$=0.5 is a compromise to achieve a good ratio between anomaly and normal events in the central region.

## Appendix B: A different Physics Example: di-jet and di-photon final state

In this section we use AA in a different final state where new physics is often searched for, the two forward jets and a central diphoton state. Instead of images of fat jets, we apply the AA method to a different input: a dataset made of events characterised by a list of values for kinematical variables in the $jj\gamma\gamma$ final state. For the normal datasets we consider two dominant production processes for the Higgs boson in the Standard Model i.e. gluon fusion and vector-boson fusion, where the Higgs decays to two photons. As anomaly examples, we consider three cases [7]. First, a ZH case, which is another production mode of the Higgs and where the Z decays to $jj$. The other two sources of anomaly are from compelling theories Beyond the Standard Model where an axion is produced with gauge bosons [40, 41] and a graviton, which is is a heavy spin-two state predicted in many extensions of the Standard Model. For the axion production with a gauge boson, we get two photons from the axion and two jets from the gauge boson. For the graviton production process we consider its decay to two photons and Higgs decay to the dominant channel of two b-jets.

We consider the following features:

1. $p_T$ $(j_1)$: transverse momentum of the leading jet.

2. $p_T$ $(j_2)$: transverse momentum of the sub-leading jet.

3. $\eta_{j_1}$: pseudo-rapidity of leading jet.

4. $p_T$ (leading $\gamma$): transverse momentum of the leading photon.

5. $\eta_{jj}$: pseudo-rapidity difference between leading and sub-leading jet.

6. $\eta_{j\gamma}$: pseudo-rapidity difference between leading jet and leading photon.

The distributions of these kinematic observables are shown in Fig. 13.

First we discuss these features for the normal data sets. We see that $p_T$ distributions of jets and photon are not very different for ggF and VBF processes. However pseudo-rapidity distributions for the considered combinations are comparatively different. Among the anomaly data sets, leading jet and photon $p_T$ distributions for axion and graviton benchmarks are quite different from ggF, VBF and the ZH case.

As the input for the AA algorithm is not a jet image, the architecture is now adapted to this situation. This is shown in Fig. 14.

The classifier output for the normal and anomaly data sets is shown in Fig.15. We see that a the baseline binary classifier tends to align anomalies towards one of the classes as already observed in the example in the main text. Then, we perform an anomaly awareness run with $\lambda = 0.5$ for different types of anomalies included. In the upper most figure in Fig. 16 one type of anomaly is added i.e. ZH and one sees already a displacement of the anomalies output. Further addition of the axion makes it comparatively more robust, and further improves with the three types of anomalies included in the AA run. As in the previous example of boosted phenomena, this experiment shows that Anomaly Awareness is effective at collecting anomalous events, and we could use the PDF distribution and follow the process described in the section III E to quantify anomaly detection.

## Appendix C: Anomaly Awareness for non-physics datasets

In this appendix we apply the AA algorithm to well-known non-physics benchmarks proposed in the Outlier Exposure paper [25], using as baseline tasks two digits in the MNIST dataset [42], and as anomalies CIFAR 10 [43], fMNIST [44] and nMNIST.

We see that for these non-physics datasets the behaviour is similar to the observed for fat jets, see Fig. 17. CIFAR-10 data set a ten class data set of color images. Fashion-MNIST (fMNIST) is also a 10-class data set of grayscale images. The negative MNIST (nMNIST) is generated from the MNIST data set by reversing the brightness of the images.

[7] These three processes are generated using the Higgs Effective Theory Feynrule modelhttps://feynrules.irmp.ucl.ac.be/wiki/HiggsEffectiveTheory at $\sqrt{s}$ = 13 TeV. We simulate these events at parton-level where we do not consider next order effects from multi-parton interactions and hadronization etc.

[1] Marco A.F. Pimentel, David A. Clifton, Lei Clifton, and Lionel Tarassenko. A review of novelty detection. Signal Processing, 99:215–249, 2014.

[2] Thomas Schlegl, Philipp Seeböck, Sebastian M Waldstein, Ursula Schmidt-Erfurth, and Georg Langs. Unsupervised anomaly detection with generative adversarial networks to guide marker discovery. In International conference on information processing in medical imaging, pages 146–157. Springer, 2017.

[3] Samet Akcay, Amir Atapour-Abarghouei, and Toby P Breckon. Ganomaly: Semi-supervised anomaly detection via adversarial training. In Asian conference on computer vision, pages 622–637. Springer, 2018.

[4] Yuchen Lu and Peng Xu. Anomaly detection for skin

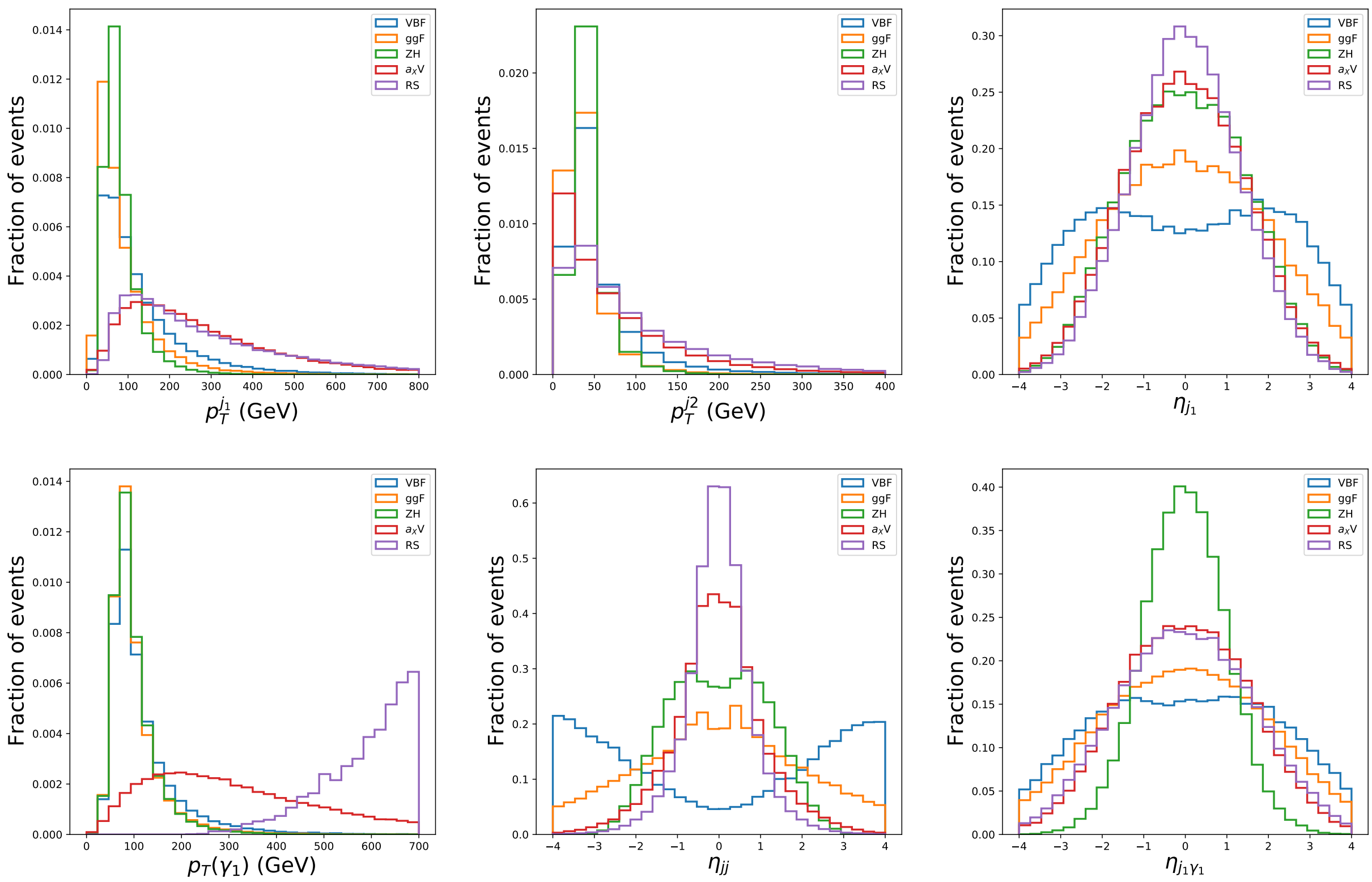

FIG. 13. Input kinematical variables of di-jet and di-photon final state considered for the second experiment.

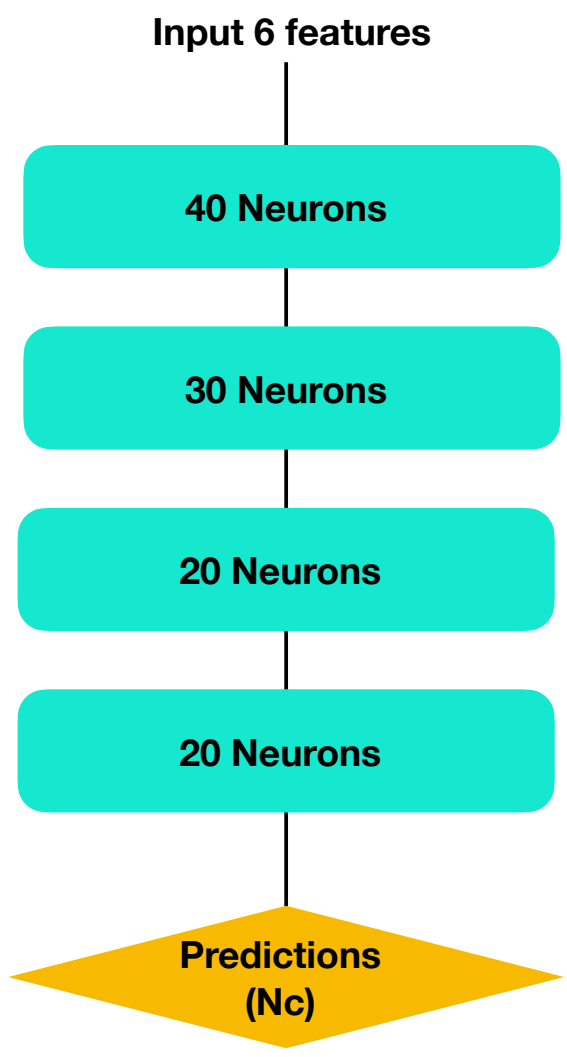

FIG. 14. Neural network architecture used for the second experiment.

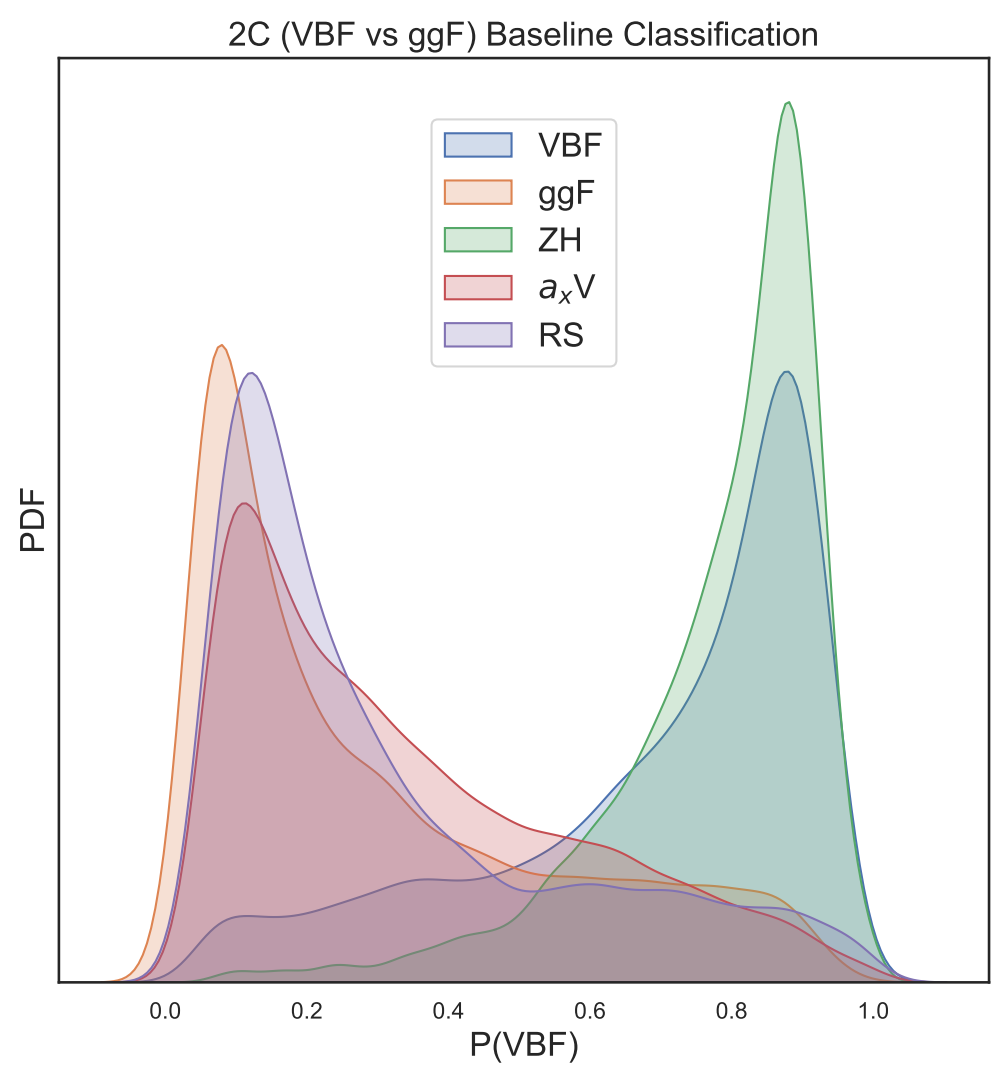

FIG. 15. Output of the neural network binary classifier for the ggF versus VBF classification.

disease images using variational autoencoder. arXiv preprint arXiv:1807.01349, 2018.
[5] Satnam Singh. Development of a machine learning based framework for visual inspection systems. 2019.
[6] Manpreet Singh Minhas and John Zelek. Anomaly detection in images. arXiv preprint arXiv:1905.13147, 2019.
[7] Eric M. Metodiev, Benjamin Nachman, and Jesse Thaler. Classification without labels: Learning from mixed samples in high energy physics. JHEP, 10:174, 2017.
[8] Jack H. Collins, Kiel Howe, and Benjamin Nachman. Anomaly Detection for Resonant New Physics with Machine Learning. Phys. Rev. Lett., 121(24):241803, 2018.
[9] Jan Hajer, Ying-Ying Li, Tao Liu, and He Wang. Novelty Detection Meets Collider Physics. Phys. Rev. D, 101(7):076015, 2020.
[10] Raffaele Tito D'Agnolo and Andrea Wulzer. Learning New Physics from a Machine. Phys. Rev. D, 99(1):015014, 2019.
[11] Andrea De Simone and Thomas Jacques. Guiding New Physics Searches with Unsupervised Learning.

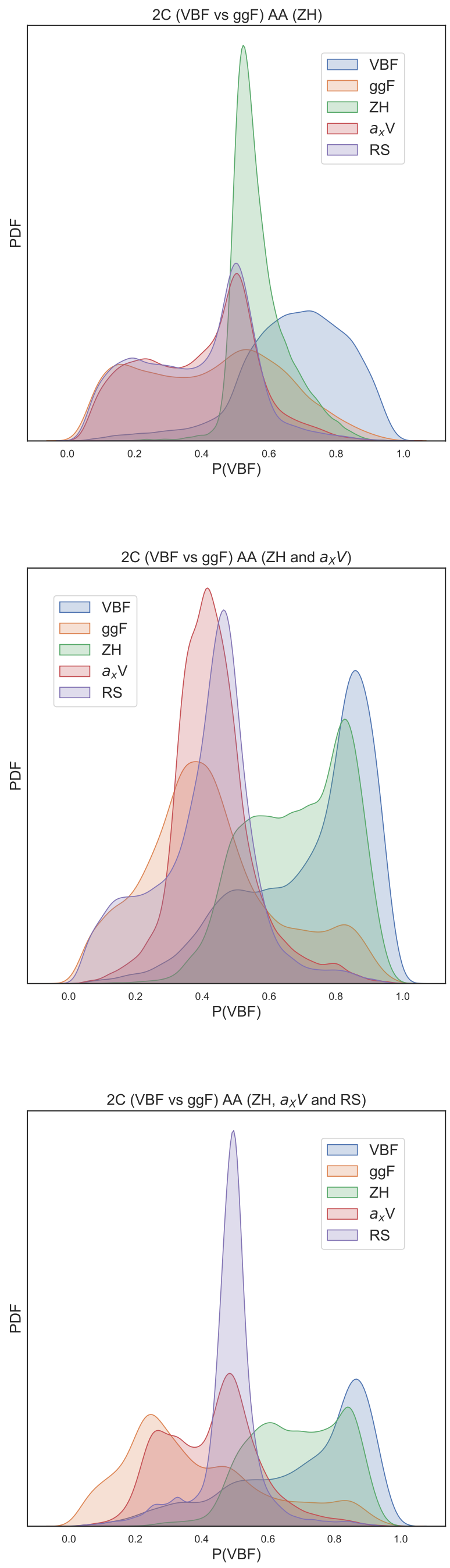

FIG. 16. Output of binary classifier after the anomaly awareness run in presence of one type, two types and three types of anomalies.

Eur. Phys. J. C, 79(4):289, 2019.

[12] Theo Heimel, Gregor Kasieczka, Tilman Plehn, and Jennifer M. Thompson. QCD or What? SciPost Phys., 6(3):030, 2019.

[13] Marco Farina, Yuichiro Nakai, and David Shih. Searching for New Physics with Deep Autoencoders. Phys. Rev. D, 101(7):075021, 2020.

[14] Olmo Cerri, Thong Q. Nguyen, Maurizio Pierini, Maria Spiropulu, and Jean-Roch Vlimant. Variational Autoencoders for New Physics Mining at the Large Hadron Collider. JHEP, 05:036, 2019.

[15] Barry M. Dillon, Darius A. Faroughy, and Jernej F. Kamenik. Uncovering latent jet substructure. Phys. Rev. D, 100(5):056002, 2019.

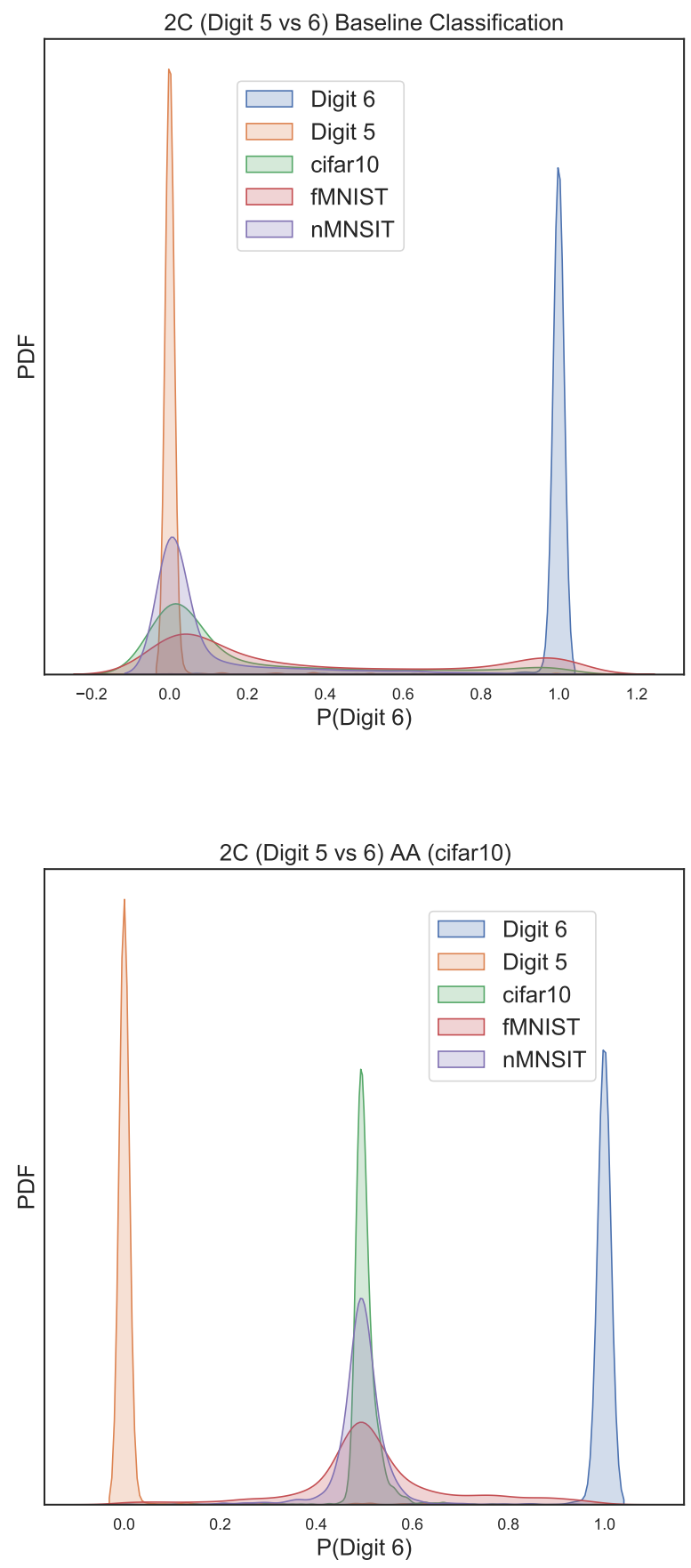

FIG. 17. AA for non particle physics data sets. We used 2 types of digits from the MNIST data set as normal examples. CIFAR 10, fMNIST and nMNIST data sets are considered as anomalies.

[16] Tuhin S. Roy and Aravind H. Vijay. A robust anomaly finder based on autoencoder. 3 2019.

[17] Andrew Blance, Michael Spannowsky, and Philip Waite. Adversarially-trained autoencoders for robust unsupervised new physics searches. JHEP, 10:047, 2019.

[18] Raffaele Tito D'Agnolo, Gaia Grosso, Maurizio Pierini, Andrea Wulzer, and Marco Zanetti. Learning Multivariate New Physics. 12 2019.

[19] Anders Andreassen, Benjamin Nachman, and David Shih. Simulation Assisted Likelihood-free Anomaly Detection. Phys. Rev. D, 101(9):095004, 2020.

[20] Benjamin Nachman and David Shih. Anomaly Detection with Density Estimation. Phys. Rev. D, 101:075042, 2020.

[21] Oz Amram and Cristina Mantilla Suarez. Tag N' Train: A Technique to Train Improved Classifiers on Unlabeled Data. 2 2020.

[22] M. Crispim Romao, N.F. Castro, and R. Pedro. Finding New Physics without learning about it: Anomaly Detection as a tool for Searches at Colliders. 6 2020.

[23] Taoli Cheng, Jean-FranÃ§ois Arguin, Julien Leissner-Martin, Jacinthe Pilette, and Tobias Golling. Variational Autoencoders for Anomalous Jet Tagging. 7 2020.

[24] Alexander Radovic, Mike Williams, David Rousseau, Michael Kagan, Daniele Bonacorsi, Alexander Himmel, Adam Aurisano, Kazuhiro Terao, and Taritree Wongjirad. Machine learning at the energy and intensity frontiers of particle physics. Nature,

560(7716):41–48, 2018.
[25] Dan Hendrycks, Mantas Mazeika, and Thomas Dietterich. Deep anomaly detection with outlier exposure. arXiv preprint arXiv:1812.04606, 2018.
[26] Felipe F. Freitas, Charanjit K. Khosa, and Veronica Sanz. Exploring the standard model EFT in VH production with machine learning. Phys. Rev. D, 100(3):035040, 2019.
[27] Celine Degrande, Benjamin Fuks, Kentarou Mawatari, Ken Mimasu, and Veronica Sanz. Electroweak Higgs boson production in the standard model effective field theory beyond leading order in QCD. Eur. Phys. J. C, 77(4):262, 2017.
[28] John Ellis, Christopher W. Murphy, Veronica Sanz, and Tevong You. Updated Global SMEFT Fit to Higgs, Diboson and Electroweak Data. JHEP, 06:146, 2018.
[29] Johan Alwall, Michel Herquet, Fabio Maltoni, Olivier Mattelaer, and Tim Stelzer. MadGraph 5 : Going Beyond. JHEP, 06:128, 2011.
[30] J. Alwall, R. Frederix, S. Frixione, V. Hirschi, F. Maltoni, O. Mattelaer, H. S. Shao, T. Stelzer, P. Torrielli, and M. Zaro. The automated computation of tree-level and next-to-leading order differential cross sections, and their matching to parton shower simulations. JHEP, 07:079, 2014.
[31] Torbjorn Sjostrand, Stephen Mrenna, and Peter Z. Skands. PYTHIA 6.4 Physics and Manual. JHEP, 05:026, 2006.
[32] Torbjorn Sjostrand, Stephen Mrenna, and Peter Z. Skands. A Brief Introduction to PYTHIA 8.1. Comput. Phys. Commun., 178:852–867, 2008.
[33] Sebastian Macaluso and David Shih. Pulling Out All the Tops with Computer Vision and Deep Learning. JHEP, 10:121, 2018.
[34] Gregor Kasieczka, Tilman Plehn, Michael Russell, and Torben Schell. Deep-learning Top Taggers or The End of QCD? JHEP, 05:006, 2017.
[35] J.A. Aguilar-Saavedra, Jack H. Collins, and Rashmish K. Mishra. A generic anti-QCD jet tagger. JHEP, 11:163, 2017.
[36] J.A. Aguilar-Saavedra and B. Zaldivar. Jet tagging made easy. Eur. Phys. J. C, 80(6):530, 2020.
[37] Daniele Barducci, Alexander Belyaev, Aoife K. M. Bharucha, Werner Porod, and Veronica Sanz. Uncovering Natural Supersymmetry via the interplay between the LHC and Direct Dark Matter Detection. JHEP, 07:066, 2015.
[38] Glen Cowan, Kyle Cranmer, Eilam Gross, and Ofer Vitells. Asymptotic formulae for likelihood-based tests of new physics. Eur. Phys. J. C, 71:1554, 2011. [Erratum: Eur.Phys.J.C 73, 2501 (2013)].
[39] Charanjit K. Khosa, Lucy Mars, Joel Richards, and Veronica Sanz. Convolutional Neural Networks for Direct Detection of Dark Matter. 11 2019.
[40] I. Brivio, M. B. Gavela, L. Merlo, K. Mimasu, J. M. No, R. del Rey, and V. Sanz. ALPs Effective Field Theory and Collider Signatures. Eur. Phys. J. C, 77(8):572, 2017.
[41] M. B. Gavela, J. M. No, V. Sanz, and J. F. de Trocóniz. Nonresonant Searches for Axionlike Particles at the LHC. Phys. Rev. Lett., 124(5):051802, 2020.
[42] Yann LeCun, Léon Bottou, Yoshua Bengio, and Patrick Haffner. Gradient-based learning applied to document recognition. Proceedings of the IEEE, 86(11):2278–2324, 1998.
[43] Alex Krizhevsky. Learning multiple layers of features from tiny images. 2009.
[44] Han Xiao, Kashif Rasul, and Roland Vollgraf. Fashion-mnist: a novel image dataset for benchmarking machine learning algorithms. 2017.