# Peer review of "Anomaly Awareness"

_SciPost Physics_

## Round 1 · Referee Report · Anonymous · 2021-11-22

Report
This paper presents a new method for anomaly detection called "Anomaly Awareness" and applies it to the detection of physics beyond the Standard Model at the LHC.
I have some general comments and then some specific ones.
General
-- The paper is written in a very strange style, as if it was pitched to a much broader non-physics audience (ML and CS people?) originally. This style does not seem suitable or appropriate for SciPost Physics. Given that the audience of the paper is high energy physicists, the authors could greatly streamline their paper (and thereby greatly improve it) by removing many unnecessary explanations.
-- The paper is written in an extremely unclear and confused way. That made it very hard for this referee to follow what was going on. I would strongly recommend that the authors go through the paper thoroughly, line by line, and make sure every single step is clearly explained and every symbol in every plot and equation is fully and clearly defined.
Requested changes
Specific
1. The definition and motivation of the anomaly awareness loss term was not at all clear to me. How does the new term "distribute the anomalous samples uniformly across the classes"? What does that even mean? What is equation (1) really trying to accomplish? What are p_1, p_2, why don't they sum to one? Given that this is the key component of the new algorithm, the authors should spend a lot more time making it clear what this loss term is attempting to do.
2. Regarding footnote 4, I gather the authors did not run a detector simulation (eg Delphes)? They should justify this choice (ie why they don't think it will matter much for their results).
3. In section III.E, it was not clear to me what anomaly epsilon_{An} referred to. Is this for the EFT anomaly?
4. In section III.E, I ask that the authors calculate and plot a dimensionless measure of significance improvement, ie the Significance Improvement Characteristic (SIC) which is e_signal/sqrt{e_background}. In their case, the background is heterogeneous (both QCD and ttbar are included), but the authors can still define an effective background efficiency after a cut on the classifier output. Showing the SIC will much more clearly indicate the value which their new anomaly detection adds.
5. In Fig 11 it should be possible to plot a baseline cross section that would reach 5sigma nominal significance without doing any anomaly detection at all. This would provide another useful point of comparison showing the value of the AA method.
6. As far as I can tell, the authors do not discuss background estimation at all in this paper. Enhancing signal over SM background is far from sufficient for establishing the discovery of new physics at the LHC; one must also have an accurate prediction for the background. The authors should comment on how they envision their method could be combined with background estimation to actually achieve a discovery of new physics at the LHC someday.
Author: Charanjit Kaur Khosa on 2022-04-19 [id 2394]
(in reply to Report 1 on 2021-11-22)We thank the referee for the careful reading of the manuscript and for their constructive criticisms. We agree that the paper writing was aimed at a too general level. We have heavily edited these sections.
Reply: No, $p_1$ and $p_2$ do not sum up to one because they are not the softmax scores. Now we have clarified this point further in the text. Please see Eq. 1 and Eq. 2 and text before those.
Reply: We agree that detector effects are important to understand the true performance of the method for new physics searches. In this paper, we propose the method and assess its performance on a simple simulation. Tools to implement detector effects and more sophisticated theoretical simulations already exist in the field. The development of novel machine learning learning methods is our priority at the moment, which one can easily patch up with the existing set-ups. As mentioned in the paper, we intend to perform such a study in the near future.
Reply: $\epsilon_{An}$ refers to the anomaly selection efficiency, and in the particular example from Fig 10 it is EFT efficiency. We have added a sentence around Eq 5 to clarify this point.
Reply: We thank the referee for this suggestion. We have now re-defined the quantity $R$ to make it dimensionless and also changed the plot in Fig 10 accordingly.
Reply: We have now added the value of cross section one would exclude without using AA in the discussion around Fig 11.
Reply: We agree that this is an important point in building an analysis pipeline using Machine Learning approaches. However, as we stressed earlier in the report as well in the paper, in this paper we are focusing on developing a method for new physics searches. Precise background estimation is something even more generic, and in this paper we have simulated the dominant backgrounds and computed their approximate cross-sections to build the normal or baseline classification of the algorithm. If one were to apply this in a realistic LHC situation, the experimental collaboration would be able to use their pipeline for e.g. fat jets, and train the baseline classification using their more sophisticated tools. As far as this method is concerned here the idea is to have a good classification of the background events to develop the AA procedure.
We hope that the referee finds satisfactory the work we have done to improve the paper and agrees with its publication.

---

## Editorial Decision

submission_&_refereeing_history